# Comparison of Fine-Needle Aspiration Cytopathology with Histopathological Examination of the Thyroid Gland in Patients Undergoing Elective Thyroid Surgery: Do We Still Need Fine-Needle Aspiration Cytopathology?

**DOI:** 10.3390/diagnostics14030236

**Published:** 2024-01-23

**Authors:** Oskar Gąsiorowski, Jerzy Leszczyński, Joanna Kaszczewska, Kamil Stępkowski, Piotr Kaszczewski, Maksymilian Baryła, Zbigniew Gałązka

**Affiliations:** Department of General, Vascular, Endocrine and Transplant Surgery, Medical University of Warsaw, ul. Żwirki i Wigury 61, 02-091 Warsaw, Poland; joanna.kaszczewska@gmail.com (J.K.); kamil.stepkowski@gmail.com (K.S.); piokasz1@gmail.com (P.K.); maksymilian.baryla@wum.edu.pl (M.B.); zbigniew.galazka@wum.edu.pl (Z.G.)

**Keywords:** fine-needle aspiration cytology (FNAC), thyroid nodules, thyroid cancer, Bethesda classification

## Abstract

Background: The thyroid gland is responsible for various functions, but it is susceptible to pathologies. The gold standard for preliminarily diagnosing thyroid abnormalities is fine-needle aspiration cytology (FNAC), although it has some limitations; thus, postoperative histopathological examination confirms the diagnosis. The aim of the present study was to compare preoperative FNAC results with postoperative histopathological examination. Methods: This study is a retrospective study based on FNAC and postoperative histopathology examination, which were compared and analyzed. Results: This study included 344 patients between 18 and 86 years old (mean age: 53.06 ± 13.89), comprising 274 females and 70 males (mean ages 52.72 ± 13.86 and 54.39 ± 14.05, respectively) with a 3.9:1 female-to-male ratio. Statistical significance between the FNAC and histopathology results was observed (*p* = 0.0000), and 86 (25.00%) patients were found to have been diagnosed incorrectly based on FNAC. The sensitivity of FNAC was 92.31%, and its specificity was 82.08%, with positive and negative predictive values of 68.57% and 96.08%, respectively. Conclusions: Due to many factors, FNAC may lead to over- or under-diagnosis, increasing the chances of complications associated with the selected treatment. However, we do not have any other more accurate tools; therefore, FNAC should still remain as the gold standard of preliminary examination.

## 1. Introduction

The thyroid gland is responsible for various functions, including cardiac regulation, e.g., cardiac output, stroke volume, growth, development, and metabolic homeostasis, in the normal state. The T3 hormone is responsible for fetal growth and central nervous system stimulation, improving responsiveness and alertness. The thyroid also plays a key role in reproductive system regulation in both sexes [1].

However, the thyroid can be involved in many pathological conditions. The most common are goiters (nodular or diffuse), hyperthyroidism (e.g., Graves–Basedow’s disease), hypothyroidism, thyroiditis (e.g., Hashimoto’s disease), and neoplasms.

Goiters are a common endocrine disorder affecting approximately 300 million people worldwide, with more than half of those affected being unaware of their condition [2].

One of the main causes of a goiter is iodine deficiency [3,4], defined by the World Health Organization (WHO) as a population mean urinary iodine (UI) excretion of less than 100 μg per liter [5]. When more than 5% of prepubertal (6–12 years) school-aged children in a given population have a goiter due to iodine deficiency, this is referred to as endemic goiter. However, excess iodine can also lead to the formation of goiters.

The incidence of spontaneous hypothyroidism ranges between 1% and 2% in iodine-enriched communities. It is ten times more common in women than in men and affects mostly older women. Its prevalence varies between 0.6 and 12 per 1000 in women and between 1.3 and 4.0 per 1000 in men, and it is generally higher in older populations. Its incidence is lower in iodine-deficient areas. The average annual incidence of spontaneous hypothyroidism during a 20-year follow-up period is 3.5 per 1000 and 0.6 per 1000 in surviving women and men, respectively [6,7,8]. Similarly, gender-related differences are observed in iodine-sufficient communities. The prevalence in women ranges from 0.5 to 2% and is ten times higher than in men. Data concerning elderly populations demonstrate a prevalence of 0.4 and 2.0%, with a higher percentage in iodine-deficient areas. The incidence data available for overt hyperthyroidism in men and women from large population studies are comparable, at 0.4 per 1000 women and 0.1 per 1000 men, but the age-specific incidence varies considerably [6,8,9].

Another condition related to the endocrine function of the thyroid gland is thyroiditis. This is defined as increased serum concentrations of thyroid antibodies such as anti-thyroid peroxidase (microsomal) (TPOAb) and anti-thyroglobulin (TGAb). Early post-mortem studies have reported histological evidence of chronic autoimmune thyroiditis in 27% of adult women, with an increasing incidence in those over 50 years old and in 7% of adult men. Diffuse changes were diagnosed in 5% of women and 1% of men [8].

Thyroid cancer is the most common endocrine neoplasm, accounting for 2.1% of all new malignancies diagnosed annually worldwide (excluding skin cancer and carcinoma in situ). The annual incidence of thyroid cancer has been increasing over the last few decades. It has almost tripled from 4.9 per 100,000 in 1975 to 14.3 per 100,000 in 2009 [6,10,11].

The five most common types of thyroid neoplasia are papillary carcinoma, follicular carcinoma, medullary carcinoma, anaplastic carcinoma, and lymphoma. Papillary and follicular carcinomas are also known as differentiated thyroid cancer (DTC). DTC accounts for more than 90% of all thyroid malignancies. Papillary thyroid cancer (PTC) accounts for about 80% of thyroid cancers in the US, while follicular thyroid cancer (FTC) accounts for about 10% of cases [6,12,13].

Medullary thyroid cancer accounts for about 3–5% of all thyroid malignancies in the US. However, it is responsible for 15% of all thyroid cancer deaths [6,14].

Anaplastic thyroid cancer is a rare tumor, accounting for less than 2% of thyroid malignancies. It is one of the most aggressive solid malignancies in humans, similar to pancreatic tumors or acute lymphoblastic leukemia, with a median survival of less than 6 months post-diagnosis and a 1-year survival rate of 20% [6,15,16,17].

An overwhelming majority of lymphomas affecting the thyroid gland are non-Hodgkin’s lymphomas. With an annual incidence of approximately 2.1 cases per million people, Hodgkin’s lymphomas account for less than 2% of all thyroid cancers [6,18,19,20].

A variety of diagnostic tests such as ultrasonography, nuclear scans, and fine-needle aspiration cytology (FNAC) are used to evaluate the entire thyroid gland and suspicious masses found in clinical examinations. Ultrasonography (USG) remains a basic tool in the diagnosis and monitoring of thyroid diseases. Following accurate histology and high-quality USG, FNAC can be performed. Currently, FNAC is considered the gold standard in the investigation of suspected thyroid nodules. Thyroid scintigraphy, based on 99mTc pertechnetate or 123I, is used to assess thyroid function, image autonomously functioning thyroid nodules, and evaluate the extent of thyroid cancer [21].

FNAC is a simple, inexpensive, easily repeatable, and quick procedure that can be performed in an outpatient setting. Important factors for reliable testing are a representative sample of the nodule and the presence of an experienced cytologist to interpret the results. FNAC is often recommended as the next step in the diagnosis of thyroid nodules after USG [22].

The first attempts at diagnosing thyroid nodules via biopsy date back to 1930, when Martin and Ellis reported an 18-gauge-needle aspiration technique. Further attempts include a needle biopsy with Silverman or Tru-Cut needles. None of these methods gained acceptance due to fears of malignant cells spreading in the needle track and complications [23,24].

The presently used FNAC was developed in Sweden in 1960 to assess the malignant nature of thyroid nodules [25] and is still the gold standard in pre-diagnostic investigations [26].

In order to standardize reporting and describe the results of thyroid cytopathology, The Bethesda System for Reporting Thyroid Cytopathology (TBSRTC) was introduced in 2010. The second edition of the TBSRTC was published in 2017. Both became widely accepted, facilitating clear communication between the cytopathologist and referring physician and a clear interpretation of the results. In 2023, a new, refined TBSRCT was introduced. In comparison with the previous classification, the 2023 system simplifies alternative names for three of the diagnostic categories, recommending a single designation for each category and discontinuing the previously used terms “unsatisfactory”, “follicular lesion of undetermined significance”, and “suspicious for a follicular neoplasm”.

In the newest 2023 edition of the TBCRTC, the following names for each of the SIC categories are recommended: (I) nondiagnostic; (II) benign; (III) atypia of undetermined significance (AUS); (IV) follicular neoplasm; (V) suspicious for malignancy (SFM); and (VI) malignant [27].

The sensitivity of FNAC ranges from 55% to 98% and the specificity from 73% to 100% [28,29].

Despite its high sensitivity and specificity, FNAC has limitations related to specimen adequacy, sampling technique, skill in performing the aspiration, interpretation of the aspirate, and overlapping cytological features between benign and malignant neoplasms. Histopathological examination of the surgically removed thyroid gland is the confirmatory diagnostic technique used to determine its pathology [30].

This study was designed to compare preoperative FNAC results with the postoperative histopathological evaluation of a surgically removed thyroid gland.

## 2. Materials and Methods

This retrospective study was conducted in the Department of General, Vascular, Endocrine and Transplant Surgery to compare the accuracy of FNAC with histopathological results in patients undergoing thyroidectomy.

### 2.1. Inclusion Criteria

The inclusion criteria were adult patients (older than 18 years) of either sex presenting indications for thyroid surgery according to The Bethesda System for Reporting Thyroid Cytopathology and the following clinical signs:A III–VI category in The Bethesda System for Reporting Thyroid Cytopathology (TBSRTC).Second category of TBSRTC, which cannot be treated conservatively.Dyspnea or pressure of the trachea.Dysphagia or esophagus compression.Uncontrolled hyperthyroidism, which cannot be treated conservatively.Retrosternal goiters.

### 2.2. Exclusion Criteria

The exclusion criteria were as follows:Patients unfit for surgery.Patients with II category of TBSRTC (the Bethesda scale) who could be treated conservatively—EU-TIRADS 1–2.Patients not consenting to participate in the study.Patients not consenting to surgery.

### 2.3. Methodology

Prior to elective surgical treatment, patients underwent endocrinological diagnostics. Following the preoperative physical examination, assessment of laboratory tests, ultrasonography, and FNAC, all patients with positive inclusion criteria were referred to elective surgery.

Thyroid specimens were preserved in 10% buffered formalin after surgery for further research. Subsequently, the specimens were examined by a histopathologist.

FNAC and histopathology reports were analyzed and correlated.

Statistical analysis was performed using Statistica 13 (StatSoft Polska Sp. z.o.o., Krakow, Poland) and the chi-squared test.

All results were classified with The Bethesda System for Reporting Thyroid Cytopathology (TBSRTC) (Table 1) [27,31].

## 3. Results

The age of patients ranged from 18 to 86 years.

This study included 344 patients (mean age of group: 53.06 ± 13.89), 274 female (79.65% of the study group; mean age: 52.72 ± 13.86) and 70 male (20.35% of the study group; mean age: 54.39 ± 14.05), with a 3.9:1 female-to-male ratio. The highest number of patients was in the age group of 41–50 years, i.e., 24.42%, followed by 61–70 years, i.e., 22.97% (Figure 1).

The number of patients undergoing FNAB in our study was 344. The first category was observed in 1 patient (0.29%), the second category in 204 (59.30%), the third in 46 (13.37%), the fourth in 44 (12.79%), the fifth in 26 (7.56%), and the sixth in 23 (6.69%) (Figure 2).

Out of 344 specimens in histopathological examination (HPE), 280 (81.40%) were benign lesions and 64 (18.60%) were malignant (Figure 3 and Table 2). The most common lesions were goiters (both adenomatous and colloid), being found in 173 patients (50.29% all of cases). Behind them were autoimmune thyroid disorders with 71 (20.64% all of lesions) cases (Figure 4). The most common malignant lesion was papillary carcinoma, with fifty-eight (16.86%) cases, followed by follicular, medullary, and oncocytoma (Hürthle) cancer with two (0.58%) cases (Figure 4). There were no anaplastic cancers or lymphoma in our study.

The benign-to-malignant ratio was 4.38:1.

HPE revealed 173 cases of goiters (adenomatous and colloid), which were most common in the category of benign lesions (280 cases) with 61.79%. This was followed by AITD with 71 cases (25.36%). Next were follicular adenoma with nineteen (6.79%), oncocytoma with nine (3.21%), AUS with seven (2.50%), and paraganglioma with one case (0.36%) (Figure 5).

HPE revealed 64 malignant lesions. The most often recognized was papillary cancer with fifty-eight cases (90.63%), followed by follicular cancer with two cases (3.13%), medullary cancer with two cases (3.13%), and oncocytoma (formerly Hürthle) cancer with two cases (3.13%). We found no anaplastic cancer or lymphoma of the thyroid gland in our study (Figure 5).

The sensitivity of FNAC was 92.31%, while its specificity was 82.08%. The positive predictive value was 68.57%, while the negative predictive value was 96.08%. There were 12.5% false positives and 2.04% false negatives. The accuracy was 84.88%.

After statistical analysis, the significance between FNAC and histopathology was *p* = 0.0000. The correlation between histopathology and gender was almost clinically significant (*p* = 0.07942), but there was no statistical significance between FNAB and gender *p* = 0.32224. There was no statistical significance between FNAC and age (*p* = 0.53671).

There was no clinical correlation between histopathology and age (*p* = 0.12242).

Incorrect diagnoses based on FNAC were observed in 86 out of 344 patients (25.00%), while correct diagnoses based on FNAC were observed in 258 out of 344 patients (75.00%).

Our study showed an increased TSBRTC level based on FNAC in 43 patients (12.50%), which means that 43 out of 344 patients were qualified higher in HPE than they should have been following FNAC.

However, for 43 of 344 patients (12.50%), the decreased Bethesda scale level based on FNAC means that these patients were qualified lower than they should be.

HPE revealed that a change in classification from category II to IV was present in 14 (4.07%) patients; from category II to VI in 6 (1.74%) patients; from category III to IV in 10 (2.91%) patients; from category III to VI in 5 (1.45%) patients; and from category IV to VI in 6 (1.74%) patients.

## 4. Discussion

In the study conducted by Surriah M.H. et. al., females 113/134 (84.3%) were more frequently affected than males 21/134 (15.7%) by any type of thyroid disease [32].

In addition, Singh P. et. al. had 13 males (18.66%) and 57 females (81.42%) in their study, with a female-to-male ratio of 4.3:1 [3].

In the study by Machała E. et. al., among 1262 patients who made up the study population and were between the ages of 18 and 82 (mean age 50 years), there were 976 (77.34%) females and 286 (22.66%) males, with a female-to-male ratio of 3.4:1 [33].

In our study, there were 274 (79.65%) females and 70 (20.35%) males, with a female-to-male ratio of 3.9:1, indicating that our results are comparable to others.

The age of the patients in Singh P. et. al. ranged from 19 to 75 years. The highest number of patients was in the 31–40-years age group (26%). The mean age of the study population was 38.99 years (SD 11.018), with a range of 19–75 years [3].

In the study by Machała E. et al., the highest number of cases was in the age group 41–50 years (25.6%), followed by the age groups 51–60 years (22.3%) and 31–40 years (22.2%) [33].

In our study, the highest number of patients was in the 41–50-years group with a percentage of 24.42%, followed by 61–70 years with 22.97%.

The mean age of our study group was 53.06 ± 13.89 years. The mean age of women and men was 52.72 ± 13.86 and 54.39 ± 14.05, respectively.

The diagnosis based on FNAC was correlated with histopathology and compared with other findings.

In the study by Singh P et al., the authors showed that FNAC had a sensitivity of 83.3%, specificity of 100%, positive predictive value of 100%, and negative predictive value of 96.7%. However, the overall accuracy in this study was found to be 95.71%. The results were found to be statistically significant (*p* < 0.05) [3].

Sharma C. conducted a study in which the false positive and false negative rates were 1.9% and 10.5%, respectively. The sensitivity and specificity were 89.5% and 98%, respectively. The positive predictive value was 84.6% and the negative predictive value was 98.6%. The accuracy of FNA was 97% [34].

Overall, in the results presented by Hajmanoochehri F. and Rabiee E., the sensitivity of FNAC diagnosis was 95.2%, the specificity was 68.4%, the positive predictive value was 83.3%, the negative predictive value was 89.6%, and the accuracy was 85.14% [35].

In the study by Machała E. et al., the sensitivity and specificity were 60.28% and 98.05%, respectively. The false positive rate was 1.95% and the false negative rate was 39.72%. The positive predictive value was 90.1% and the negative predictive value was 89.35%. The accuracy of FNA in differentiating benign from malignant thyroid lesions was 89.46% [33].

In our study, the sensitivity of the test was 92.31%, while the specificity was 82.08% and was statistically significant (*p* < 0.05), which is comparable with other findings. The positive predictive value was 68.57% and was lower compared to the studies by Sharma C [34], Singh P. et al. [3], Hajmanoochehri F. and Rabiee E. [35], and Machała E. et al. [33], while the negative predictive value was 96.08% and was similar.

The accuracy, referring to the differentiation of malignant lesions from benign lesions, was 84.88% and was parallel to that found in the studies of Machała E. et al. [33] and Hajmanoochehri F. and Rabiee E. [35].

Machała E. et al. reported 77.26% benign and 22.74% malignant lesions in their analysis. The ratio of malignant to benign lesions was 1:3.4. The most common benign lesion was a colloid goiter. The most common neoplasm according to histopathology was papillary carcinoma in 196 patients (68.3% of all diagnosed cancers), followed by follicular carcinoma in 51 (17.7%), medullary carcinoma in 15 (5.2%), anaplastic carcinoma in 8 (2.5%), and other in 17 (5.9%) [33].

Singh P. reported 70 cases of FNAC compared with histopathological examination (HPE). Colloid goiter was observed in 46 cases (65.71%) with FNAC and in 45 cases (64.28%) with HPE. Colloid goiters with cysts were seen in 12 cases (17.14%) with FNAC and 11 cases (15.71%) with HPE. Lymphocytic thyroiditis and adenomatous goiter were seen in one case each (1.42%) in both FNAC and biopsy. Papillary carcinoma was found in seven cases (10%) in FNAC and nine cases in HPE. Medullary carcinoma was found in two cases. Anaplastic carcinoma was found in one case in HPE [3].

Hajmanoochehri F. and Rabiee E. reported papillary carcinoma in 75% and follicular carcinoma in 7.7%. Follicular adenoma accounted for 10.9% of our cases. In addition, Hürthle cell carcinoma accounted for 5.8% of malignant cases in our study. However, previous studies have reported a lower percentage for this type of follicular carcinoma or have failed to study it as a separate malignancy type [35].

In our study, 280 out of 344 (81.40%) thyroid changes were benign. Colloid and adenomatous goiter were the most common lesions in our findings 173/344 (50.29%), followed by AITD lesions in 71 cases. Malignancy developed in 64/344 (18.60%) cases. The most common malignant lesion was papillary carcinoma (58/64, 90.62%), followed by follicular carcinoma (2/64), medullary carcinoma (2/64), and oncocytoma (formerly Hürthle cell) carcinoma (2/64) with 3.13%. No anaplastic cancers or lymphomas were found in our study. This is comparable with other findings, where the most common benign lesion was a goiter and the most common malignant lesion was papillary cancer (range 75–95% for malignant lesions).

However, from a clinical point of view, we must remember the increased percentage of malignancy in categories III and IV of the Bethesda scale, which cannot be ignored. Zahid A. et al. reported malignancy in Bethesda category III and Bethesda category IV thyroid nodules in 29.6% and 47.1% of cases, respectively [36]. Bayrak B.E. and Eruyar A.T. [37] found malignancy rates of 25% for category III and 27.6% for category IV in patients who underwent surgery. In our findings, the percentage of malignancy in categories III and IV of the Bethesda scale was 1.74%, which is not comparable with the findings of Zahid A. et al. [36] and Bayrak B.E. and Eruyar A. [37]; however, all endocrinological surgeons should remember that categories III and IV carry an increased risk of malignancy, especially category IV.

The Bethesda System for Reporting Thyroid Cytopathology (TBSRTC) is the most popular utilized classification for reporting thyroid FNAC, and categories III, IV, and V of the TBSRTC are considered grey areas of intermediate malignancy risk. The recent TBSRTC system has advocated for the use of molecular studies to identify the genetic mutations/gene rearrangements in category III or atypia of undetermined significance (AUS) in accordance with the American Thyroid Association guidelines. This would potentially identify patients for surgery, including near-total thyroidectomy or lobectomy. The most common alterations involve the B-type RAF kinase (BRAF) and RAS genes, followed by rearrangements of the RET/PTC and PAX8/PPARγ genes. The BRAF V600E mutation is present in 45–80% of classical PTC, 5–25% of follicular variants of PTC or FVPTC, 1.4% of follicular thyroid carcinomas, 5–15% of poorly differentiated thyroid carcinomas or PDC, and 10–50% of anaplastic thyroid carcinomas [38].

According to the latest TBSRTC from 2023, follicular lesions of undetermined significance (FLUS) were removed from the reporting system. At present, FNA lesions should qualify as AUS. Currently, AUS is divided into two major cytopathology types: nuclear AUS and other AUS. AUS themselves carry an increased risk of malignancy; however, nuclear atypia carry an increased risk of malignancy compared to other AUS [27].

In our studies, in reference to FNA and TBSRTC 2023, category III has been advanced to category IV or VI in HPE. Fifteen patients (4.36%) experienced a misleading diagnosis, which could lead to an underestimated probability of cancer growth. Referring to the obtained results of FNA, the following options should have been considered: repeat FNA, molecular test, diagnostic lobectomy, and surveillance. Repeating FNA no sooner than 3 months following the initial aspiration [39] may result in uncontrolled cancer growth if it is not correctly diagnosed in time, for example, in the case of inaccurate FNA examination (4.36% of cases in our study). Molecular tests should be developed to correctly recognize lesions and thus improve diagnosis. However, 100% accurate markers allowing us to distinguish nuclear AUS from other AUS have not yet been developed [38].

This field may be developed in the future and could represent the method of choice for category III regarding the decision for elective thyroid gland surgery.

However, countries without adequate financial security would not be able to perform expensive molecular tests at present. Diagnostic lobectomy has shown usefulness in the field, as patients are able to keep the functional lobe of their thyroid gland. However, for cancer detection, another surgery is required, which may put the patient at risk. Another surgery with general anesthesia, especially in those patients with comorbidities, is risky. Watchful waiting and satisfactory ambulatory medical care may represent acceptable alternatives. Nonetheless, some approaches may not be achievable in developing countries due to procedure costs and a lack of medical personnel and equipment.

## 5. Conclusions

FNAC is the most important preliminary test for diagnosing thyroid disease because it is safe, inexpensive, simple, and quick.

However, FNAC is not 100% accurate, as it has a margin of error. This inaccuracy problem is a result of human error in the collection of biopsy material.

As a result of FNAC over-diagnosis, unnecessary elective surgery is sometimes performed when conservative therapy would suffice, which may lead to postoperative complications (such as bleeding, hypoparathyroidism, and vocal cord paralysis) in people who are eligible for elective surgery. However, FNAC can also lead to under-diagnosis, resulting in clinical deterioration and inappropriate surgical timing.

Currently, however, we do not have any other more accurate tools that can allow a superior preoperative differential diagnosis of thyroid lesions. For the grey zone areas with an intermediate risk of malignancy (TBSRTC III to V), it is possible to perform DNA extraction and use a real-time polymerase chain reaction to predict malignant lesions. According to TBSRTC 2023, molecular factors responsible for nuclear AUS, which may be identified based on one simple, cost-effective molecular test, should be determined. This would aid in recognizing the increased likelihood of malignancy in conditions aside from other AUS, as the probability of malignancy is decreased in comparison to nuclear AUS. Furthermore, it would prevent the overestimated need for thyroid gland resection.

In developed countries, molecular testing to exclude malignancy and surveillance appear to be the preferred methods as a consequence of satisfactory access to advanced technology and a wide range of medical personnel. Conversely, in developing countries, especially those with greater thyroid disease morbidity, elective thyroid surgery may be preferred over surveillance due to limited primary medical care and molecular testing resources.

Therefore, histopathological examination remains the gold standard for final diagnosis; however, we should look for a minimally invasive procedure that is cost-effective and provides a higher-quality diagnosis than FNAC.

## Figures and Tables

**Figure 1 diagnostics-14-00236-f001:**
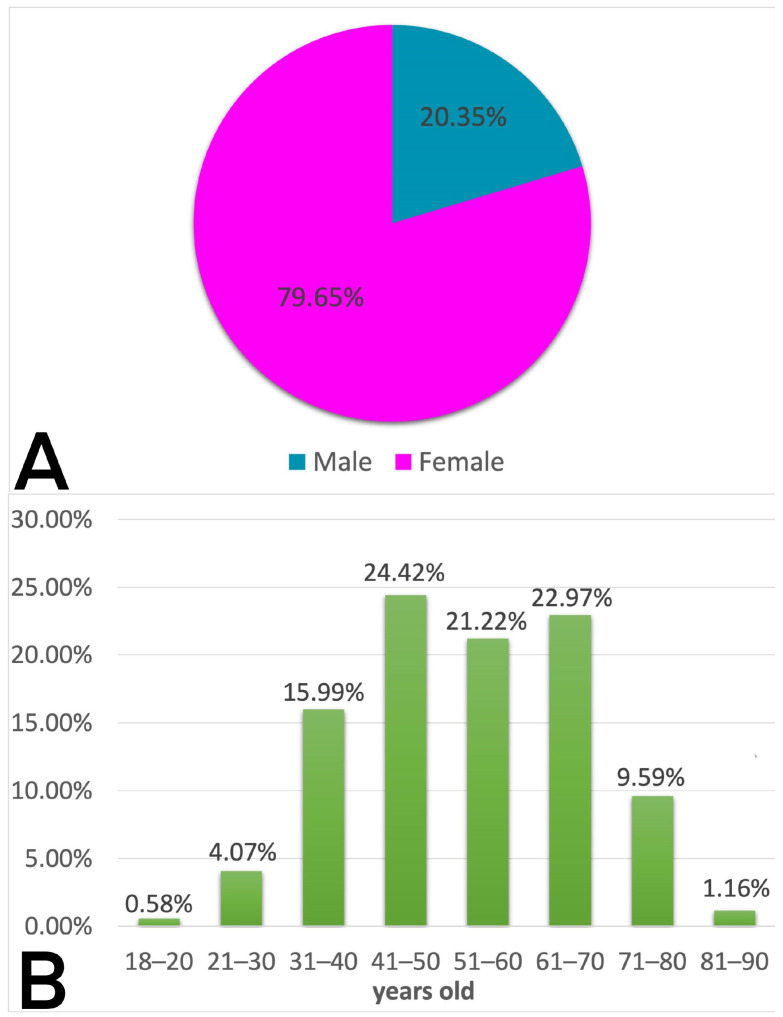
(**A**) Sex distribution in our study. (**B**) Distribution of age in study group.

**Figure 2 diagnostics-14-00236-f002:**
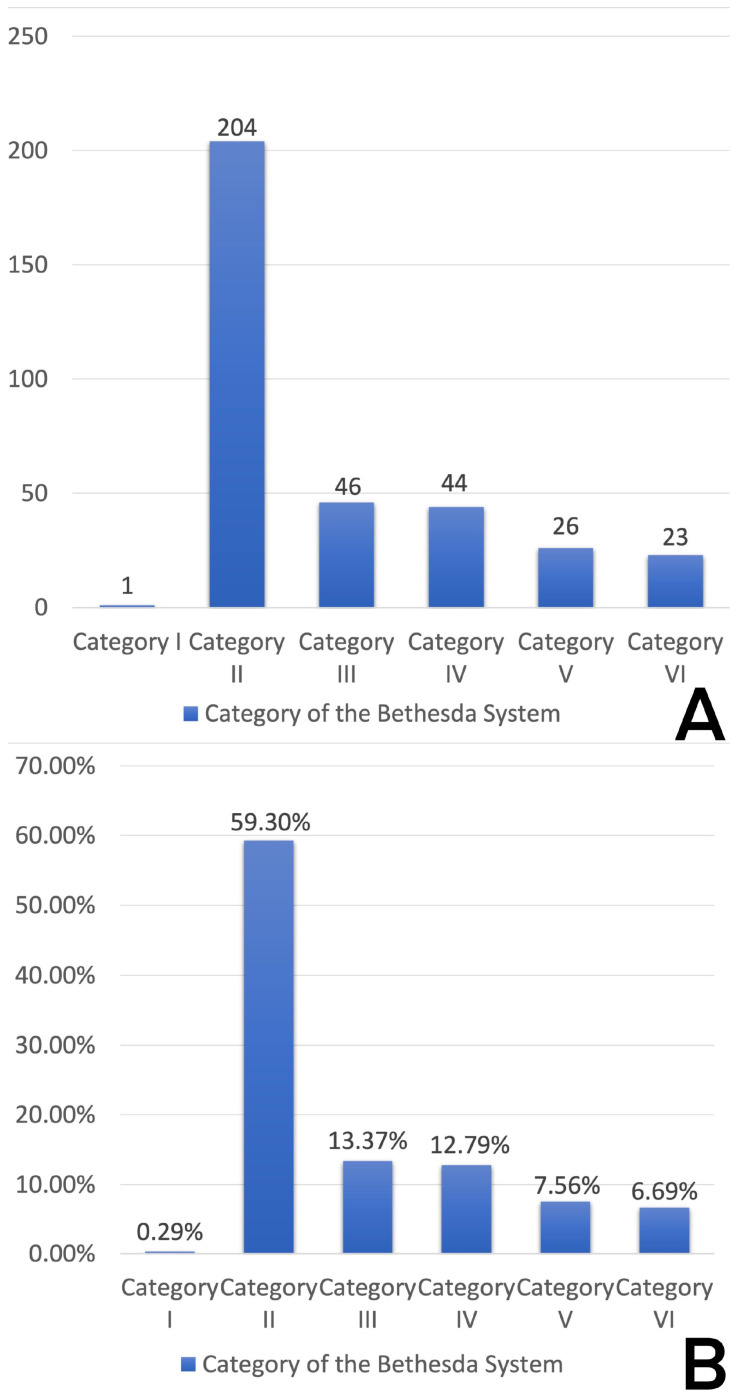
(**A**) Number distribution of The Bethesda System for Reporting Thyroid Cytopathology (TBSRTC) categories in our findings. (**B**) Percentage distribution of categories of The Bethesda System for Reporting Thyroid Cytopathology (TBSRTC) in our findings.

**Figure 3 diagnostics-14-00236-f003:**
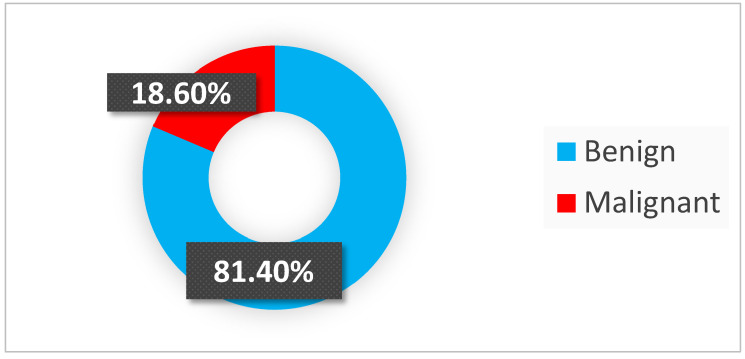
Percentages of lesion types.

**Figure 4 diagnostics-14-00236-f004:**
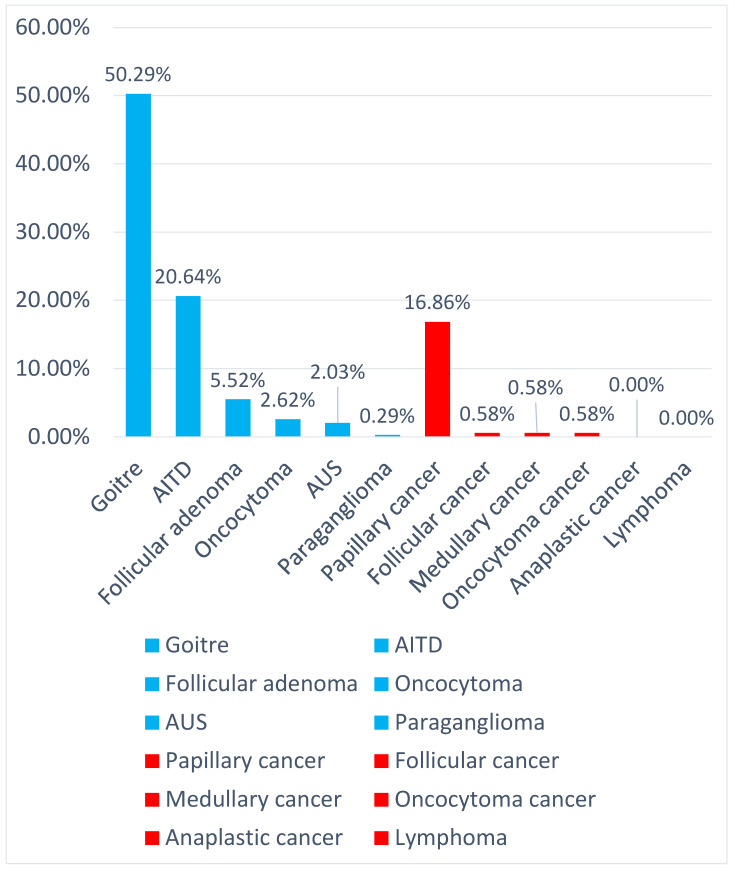
Percentage of the types of lesions in this study.

**Figure 5 diagnostics-14-00236-f005:**
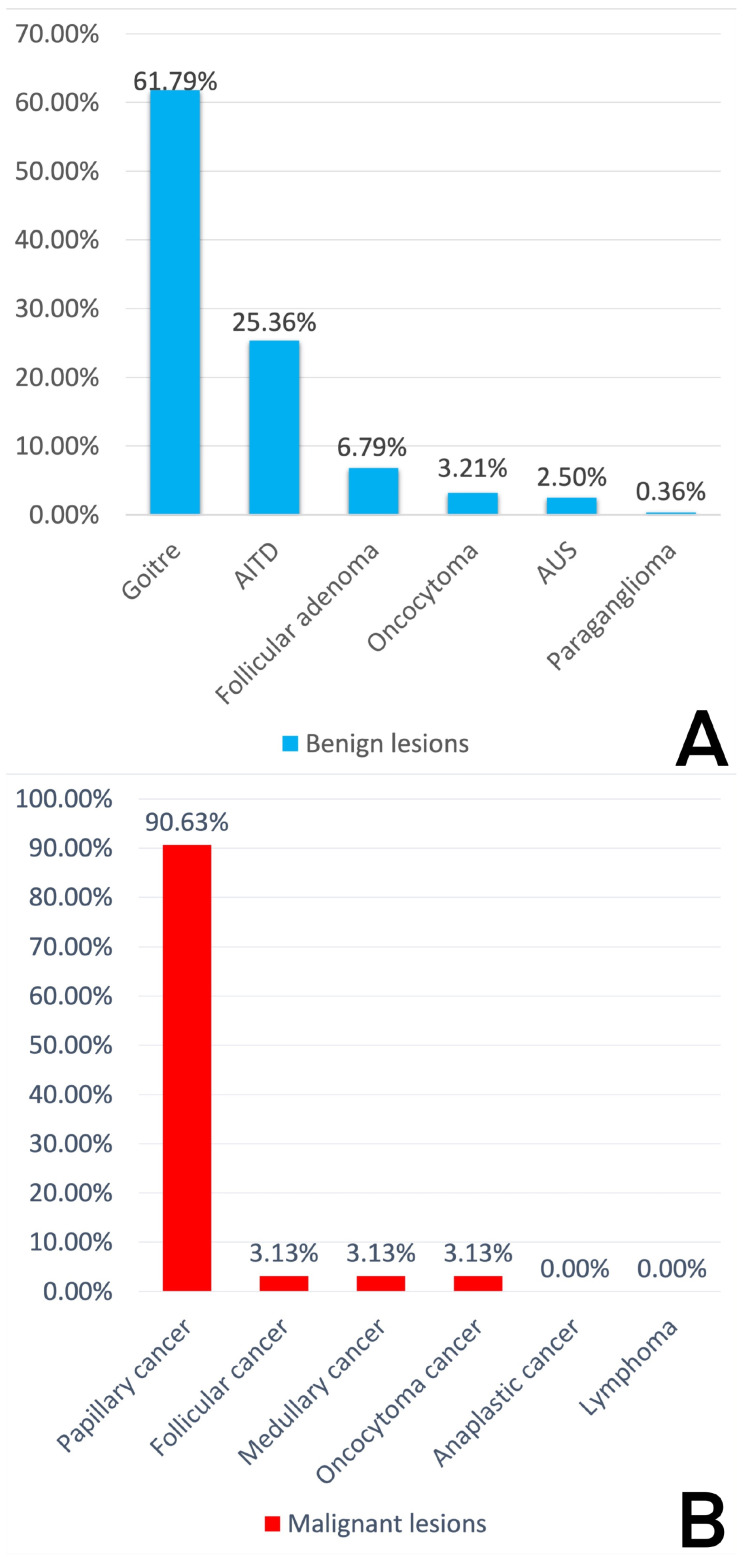
(**A**) Percentage distribution of the benign lesions. (**B**) Percentage distribution of the malignant lesions.

**Table 1 diagnostics-14-00236-t001:** The Bethesda System for Reporting Thyroid Cytopathology.

The Bethesda System for Reporting Thyroid Cytopathology	
I	Nondiagnostic or unsatisfactory
II	Benign
III	Atypia of undetermined significance (AUS)
IV	Follicular neoplasm or suspicious for a follicular neoplasm
V	Suspicious of malignancy
VI	Malignant

**Table 2 diagnostics-14-00236-t002:** FNAC and HPE results comparison.

Cytological Results According to Bethesda System		Histopathology	
		Benign	Malignant
I	1	1	0
II	204	198	6
III	46	41	5
IV	44	38	6
V	26	2	24
VI	23	0	23
TOTAL	344	280	64

## Data Availability

Data are contained within the article.

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
