# Peer review of "Comparison of Fine-Needle Aspiration Cytopathology with Histopathological Examination of the Thyroid Gland in Patients Undergoing Elective Thyroid Surgery: Do We Still Need Fine-Needle Aspiration Cytopathology?"

_diagnostics, 2024, doi:10.3390/diagnostics14030236_

Round 1

Reviewer 1 Report

Comments and Suggestions for Authors

In this study of 344 patients, the statistical significance between FNAC and histopathology results was documented. Overall, 86 (25,00%) patients were diagnosed incorrectly based on FNAC. Sensitivity and specificity of FNAC reached 92,31% and 82,08%, respectively, and with positive and negative predictive value of 68,57% and 96,08%, respectively. The authors concluded that, despite evident shortcomings, FNAC should still remain as the golden standard of preliminary examination. I agree.

Author Response

Dear Sir or Madam,

I would like to thank you for the time and effort devoted to prepare the review of our article and your agreement with our statement.

Reviewer 2 Report

Comments and Suggestions for Authors

This is a study comparing FNA results with surgical pathology results in thyroid tumors. There are many reports of these epidemiological studies, with comparable sensitivity, specificity, and other values. Although the number of cases examined in this study is large, the results are comparable to previous reports and lack any new points to discuss. In actual clinical practice, surgery may be decided based on the results of FNA, but even if the results suggest the possibility of malignancy, the patient may be followed up depending on the size of the tumor.

https://pubmed.ncbi.nlm.nih.gov/30560718/

Based on the results obtained from a number of cases, I guess the authors have to discuss the means, methods, and changes in treatment strategies to better improve the current situation and what improvements can be made in the future.

  Comments on the Quality of English Language

The section of discussion is very classical in format and limited to comparisons with other papers. In addition, there are many similarly formatted statements in English, and even if the grammer is not wrong, there is an impression that the context is insufficient. 

It would be necessary to develop an argument that would derive new considerations and improvements based on the results obtained from this study, rather than a comparison with other reports.

Author Response

Dear Sir or Madam,

 on behalf of all authors, I would like to thank you for the time and effort devoted to prepare the review of our article as well as for the constructive comments that allowed us to refine the content of our manuscript.

Round 2

Reviewer 2 Report

Comments and Suggestions for Authors

This is an epidemiological study with evaluation according to the new diagnostic criteria of TBSRTC 2023 and pathological evaluation from resected specimens, and many additional revisions clearly indicate the purpose of the study and future developments resulting from the results.

Appropriate improvements have been made in response to my review comments, and the description is considered worthy of submission.